# ABS: Automatic Bit Sharing for Model Compression

## Abstract

We present Automatic Bit Sharing (ABS) to automatically search for optimal model compression configurations (e.g., pruning ratio and bitwidth). Unlike previous works that consider model pruning and quantization separately, we seek to optimize them jointly. To deal with the resultant large designing space, we propose a novel **super-bit** model, a single-path method, to encode all candidate compression configurations, rather than maintaining separate paths for each configuration. Specifically, we first propose a novel decomposition of quantization that encapsulates all the candidate bitwidths in the search space. Starting from a low bitwidth, we sequentially consider higher bitwidths by recursively adding reassignment offsets. We then introduce learnable binary gates to encode the choice of bitwidth, including filter-wise 0-bit for pruning. By jointly training the binary gates in conjunction with network parameters, the compression configurations of each layer can be automatically determined. Our ABS brings two benefits for model compression: 1) It avoids the combinatorially large design space, with a reduced number of trainable parameters and search costs. 2) It also averts directly fitting an extremely low bit quantizer to the data, hence greatly reducing the optimization difficulty due to the non-differentiable quantization. Experiments on CIFAR-100 and ImageNet show that our methods achieve significant computational cost reduction while preserving promising performance.

## 1 Introduction

Deep neural networks (DNNs) have achieved great success in many challenging computer vision tasks, including image classification (Krizhevsky et al., 2012; He et al., 2016) and object detection (Lin et al., 2017a;b). However, a deep model usually has a large number of parameters and consumes huge amounts of computational resources, which remains great obstacles for many applications, especially on resource-limited devices with limited memory and computational resources, such as smartphones. To reduce the number of parameters and computational overhead, many methods (He et al., 2019; Zhou et al., 2016) have been proposed to conduct model compression by removing the redundancy while maintaining the performance.

In the last decades, we have witnessed a lot of model compression methods, such as network pruning (He et al., 2017; 2019) and quantization (Zhou et al., 2016; Hubara et al., 2016). Specifically, network pruning reduces the model size and computational costs by removing redundant modules while network quantization maps the full-precision values to low-precision ones. It has been shown that sequentially perform network pruning and quantization is able to get a compressed network with small model size and lower computational overhead (Han et al., 2016). However, performing pruning and quantization in a separate step may lead to sub-optimal results. For example, the best quantization strategy for the uncompressed network is not necessarily the optimal one after network pruning. Therefore, we need to consider performing pruning and quantization simultaneously.

Recently, many attempts have been made to automatically determine the compression configurations of each layer (i.e., pruning ratios, and/or bitwidths), either based on reinforcement learning (RL) (Wang et al., 2019), evolutionary search (ES) (Wang et al., 2020), Bayesian optimization (BO) (Tung & Mori, 2018) or differentiable methods (Wu et al., 2018; Dong & Yang, 2019). In particular, previous differentiable methods formulate model compression as a differentiable searching problem to explore the search space using gradient-based optimization. As shown in Figure 1(a), each candi-

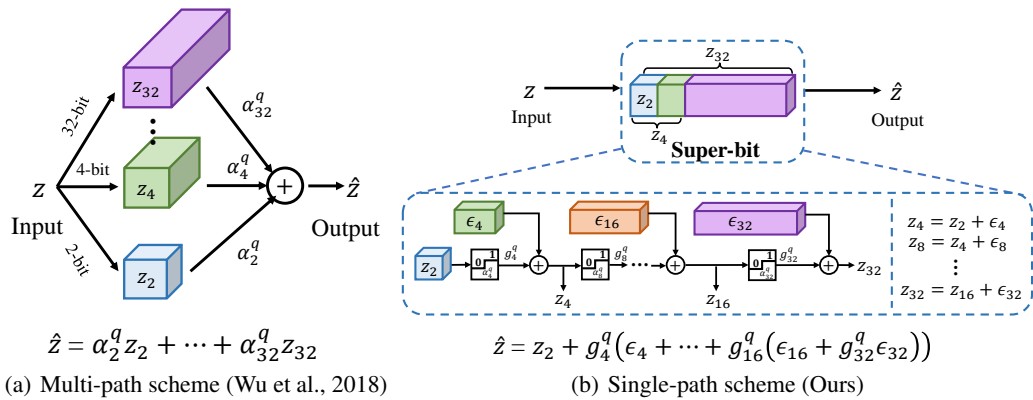

$$\hat{z} = \alpha_2^q z_2 + \cdots + \alpha_{32}^q z_{32}$$

$$\hat{z} = z_2 + g_4^q\big(\epsilon_4 + \cdots + g_{16}^q\big(\epsilon_{16} + g_{32}^q\epsilon_{32}\big)\big)$$

(a) Multi-path scheme (Wu et al., 2018)     (b) Single-path scheme (Ours)

Figure 1: Multi-path v.s. single-path compression scheme. (a) Multi-path search scheme (Wu et al., 2018): represents each candidate configuration as a separate path and formulates the compression problem as a path selection problem, which gives rise to huge numbers of trainable parameters and high computational overhead when the search space becomes combinatorially large. Here, $z_k$ is the $k$-bit quantized version of $z$ and $\alpha_k^q$ is the architecture parameters corresponding to the path of $k$-bit quantization. (b) Single-path search scheme (Ours): represents each candidate configuration as a subset of a "super-bit" and formulates the compression problem as a subset selection problem, which greatly reduces the computational costs and optimization difficulty from the discontinuity of quantization. Here, the super-bit denotes the highest bitwidth in the search space, $g_k^q$ is a binary gate that controls the decision of bitwidth, and $\epsilon_k$ is the re-assignment offset (quantized residual error).

date operation is maintained as a separate path, which leads to a huge number of trainable parameters and high computational overhead when the search space becomes combinatorially large. Moreover, due to the non-differentiable quantizer and pruning process, the optimization of heavily compressed candidate networks can be more challenging than that in the conventional search problem.

In this paper, we propose a simple yet effective model compression method named Automatic Bit Sharing (ABS) to reduce the search cost and ease the optimization for the compressed candidates. Inspired by recent single-path neural architecture search (NAS) methods (Stamoulis et al., 2019; Guo et al., 2020), the proposed ABS introduces a novel *single-path super-bit* to encode all effective bitwidths in the search space instead of formulating each candidate operation as a separate path, as shown in Figure 1(b). Specifically, we build upon the observation that the quantized values of a high bitwidth can share the ones of low bitwidths under some conditions. Therefore, we are able to decompose the quantized representation into the sum of the lowest bit quantization and a series of re-assignment offsets. We then introduce learnable binary gates to encode the choice of bitwidth, including filter-wise 0-bit for pruning. By jointly training the binary gates and network parameters, the compression ratio of each layer can be automatically determined. The proposed scheme has several advantages. First, we only need to solve the search problem as finding which subset of the super-bit to use for each layer's weights and activations rather than selecting from different paths. Second, we enforce the candidate bitwidths to share the common quantized values. Hence, we are able to optimize them jointly instead of separately, which greatly reduces the optimization difficulty from the discontinuity of discretization.

Our main contributions are summarized as follows:

- We devise a novel super-bit scheme that encapsulates multiple compression configurations in a unified single-path framework. Relying on the super-bit scheme, we further introduce learnable binary gates to determine the optimal bitwidths (including filter-wise 0-bit for pruning). The proposed ABS casts the search problem as subset selection problem, hence significantly reducing the search cost.

- We formulate the quantized representation as a gated combination of the lowest bitwidth quantization and a series of re-assignment offsets, in which we explicitly share the quantized values between different bitwidths. In this way, we enable the candidate operations to learn jointly rather than separately, hence greatly easing the optimization, especially in the non-differentiable quantization scenario.

- We evaluate our ABS on CIFAR-100 and ImageNet over various network architectures. Extensive experiments show that the proposed method achieves the state-of-the-art performance. For example, on ImageNet, our ABS compressed MobileNetV2 achieves $28.5\times$ Bit-Operation (BOP) reduction with only 0.2% performance drop on the Top-1 accuracy.

# 2 RELATED WORK

**Network quantization.** Network quantization represents the weights, activations and even gradients in low-precision to yield compact DNNs. With low-precision integers or power-of-two representations, the heavy matrix multiplications can be replaced by efficient bitwise operations, leading to much faster test-time inference and lower power consumption. To improve the quantization performance, current methods either focus on designing accurate quantizers by fitting the quantizer to the data (Jung et al., 2019; Zhang et al., 2018; Choi et al., 2018; Cai et al., 2017), or seek to approximate the gradients due to the non-differentiable discretization (Ding et al., 2019; Louizos et al., 2019; Zhuang et al., 2020). Moreover, most previous works assign the same bitwidth for all layers (Zhou et al., 2016; Zhuang et al., 2018a; 2019; Jung et al., 2019; Jin et al., 2019; Li et al., 2020; Esser et al., 2020). Though attractive for simplicity, setting a uniform precision places no guarantee on optimizing network performance since different layers have different redundancy and arithmetic intensity. Therefore, several studies proposed mixed-precision quantization (Wang et al., 2019; Dong et al., 2019; Wu et al., 2018; Uhlich et al., 2020) to set different bitwidths according to the redundancy of each layer. In this paper, based on the proposed quantization decomposition, we devise an approach that can effectively learn appropriate bitwidths for each layer through gradient-based optimization.

**NAS and pruning.** Neural architecture search (NAS) aims to automatically design efficient architectures with low model size and computational costs, either based on reinforcement learning (Pham et al., 2018; Guo et al., 2019), evolutionary search (Real et al., 2019) or gradient-based methods (Liu et al., 2019a). In particular, gradient-based NAS has gained increased popularity, where the search space can be divided into the multi-path design (Liu et al., 2019a; Cai et al., 2019) and single-path formulation (Stamoulis et al., 2019; Guo et al., 2020), depending on whether adding each operation as a separate path or not. While prevailing NAS methods optimize the network topology, the focus of this paper is to search optimal compression ratios for a given architecture. Moreover, network pruning can be treated as fine-grained NAS, which aims at removing redundant modules to accelerate the run-time inference speed, giving rise to methods based on unstructured weight pruning (Han et al., 2016; Guo et al., 2016) or structured channel pruning (He et al., 2017; Zhuang et al., 2018b; Luo et al., 2017). Based on channel pruning, our paper further takes quantization into consideration to generate more compact networks.

**AutoML for model compression.** Recently, much effort has been put into automatically determining either the optimal pruning rate (Tung & Mori, 2018; Dong & Yang, 2019; He et al., 2018), or the bitwidth (Lou et al., 2019; Cai & Vasconcelos, 2020) of each layer via hyper-parameter search, without relying on heuristics. In particular, HAQ (Wang et al., 2019) employs reinforcement learning to search bitwidth strategies with the hardware accelerator's feedback. Meta-pruning (Liu et al., 2019b) uses meta-learning to generate the weight parameters of the pruned networks and then adopts an evolutionary search algorithm to find the layer-wise sparsity for channel pruning. More recently, several studies (Wu et al., 2018; Cai & Vasconcelos, 2020) have focused on using differentiable schemes via gradient-based optimization.

**Closely related methods.** To further improve the compression ratio, several methods propose to jointly optimize pruning and quantization strategies. In particular, some works only support weight quantization (Tung & Mori, 2018; Ye et al., 2019) or use fine-grained pruning (Yang et al., 2020). However, the resultant networks cannot be implemented efficiently on edge devices. Recently, several methods (Wu et al., 2018; Wang et al., 2020; Ying et al., 2020) have been proposed to consider filter pruning, weight quantization, and activation quantization jointly. In contrast to these methods, we carefully design the compression search space by sharing the quantized values between different candidate configurations, which significantly reduces the search cost and eases the optimization. Compared with those methods that share the similarities of using quantized residual errors (Chen et al., 2010; Gong et al., 2014; Li et al., 2017b; van Baalen et al., 2020), our proposed method recursively uses quantized residual errors to decompose a quantized representation as a set of candidate bitwidths and parameterize the selection of optimal bitwidth via binary gates.

Our proposed ABS and Bayesian Bits (van Baalen et al., 2020) are developed concurrently that share a similar idea of quantization decomposition. Critically, our ABS differs from Bayesian Bits in several aspects: 1) The quantization decomposition in our methods can be extended to non-power-of-two bit widths (i.e., $b_1$ can be set to arbitrary appropriate integer values), which is a general case of the one in Bayesian Bits. 2) The optimization problems are different. Specifically, we formulate model compression as a single-path subset selection problem while Bayesian Bits casts the optimization of the binary gates to a variational inference problem that requires more relaxations and hyperparameters. 3) Our compressed models with less or comparable BOPs outperform those of Bayesian Bits by a large margin on ImageNet (See Table 2).

## 3 PROPOSED METHOD

### 3.1 PRELIMINARY: NORMALIZATION AND QUANTIZATION FUNCTION

Without loss of generality, given a convolutional layer, let $x$ and $w$ be the activations of the last layer and its weight parameters, respectively. First, for convenience, following (Choi et al., 2018; Bai et al., 2019), we can normalize $x$ and $w$ into scale $[0, 1]$ by $T_x$ and $T_w$, respectively:

$$z_x = T_x(x) = \text{clip}\left(\frac{x}{v_x}, 0, 1\right),$$ (1)

$$z_w = T_w(w) = \frac{1}{2}\left(\text{clip}\left(\frac{w}{v_w}, -1, 1\right) + 1\right),$$ (2)

where the function $\text{clip}(v, v_{\text{low}}, v_{\text{up}}) = \min(\max(v, v_{\text{low}}), v_{\text{up}})$ clips any number $v$ into the range $[v_{\text{low}}, v_{\text{up}}]$, and $v_x$ and $v_w$ are trainable quantization intervals which indicate the range of weights and activations to be quantized. Then, we can apply the following function to quantize the normalized activations and parameters, namely $z_x \in [0, 1]$ and $z_w \in [0, 1]$, to discretized ones:

$$D(z, s) = s \cdot \text{round}\left(\frac{z}{s}\right),$$ (3)

where $\text{round}(\cdot)$ returns the nearest integer of a given value and $s$ denotes the normalized step size. Typically, for $k$-bit quantization, the normalized step size $s$ can be computed by

$$s = \frac{1}{2^k - 1}.$$ (4)

After doing the $k$-bit quantization, we shall have $2^k - 1$ quantized values. Specifically, we obtain the quantization $Q(w)$ and $Q(x)$ by

$$Q(w) = T_w^{-1}(D(z_w, s)) = v_w \cdot (2 \cdot D(z_w, s) - 1),$$ (5)

$$Q(x) = T_x^{-1}(D(z_x, s)) = v_x \cdot D(z_x, s),$$ (6)

where $T_w^{-1}$ and $T_x^{-1}$ denote the inverse functions of $T_w$ and $T_x$, respectively.

### 3.2 BIT SHARING DECOMPOSITION

Previous methods consider different compression configurations as different paths and reformulate model compression as a path selection problem, which gives rise to a huge number of trainable parameters and high computational costs. In this paper, we seek to conduct filter pruning and quantization simultaneously by solving the following problem:

$$\min_{\mathbf{W}, \alpha^p, \alpha^q} \mathcal{L}\left(\mathbf{W}, \alpha^p, \alpha^q\right),$$ (7)

where $\mathcal{L}(\cdot)$ denotes some losses, and $\mathbf{W}$ is the parameters of the network. $\alpha^p$ and $\alpha^q$ are the pruning and quantization configurations, respectively. As shown in Eq. (7), we propose to encode all compression configurations in a single-path super-bit model (See Figure 1(b)). In the following, we first introduce the bit sharing decomposition and then describe how to learn for compression.

To illustrate the bit sharing decomposition, we begin with an example of 2-bit quantization for $z \in \{z_x, z_w\}$. Specifically, we consider using the following equation to quantize $z$ to 2-bit:

$$z_2 = D(z, s_2), \quad s_2 = \frac{1}{2^2 - 1},$$ (8)

where $z_2$ and $s_2$ are the quantized value and the step size of 2-bit quantization, respectively. Due to the large step size, the residual error $z - z_2 \in [-s_2/2, s_2/2]$ may be big and result in a significant performance drop. To reduce the residual error, an intuitive way is to use a smaller step size, which indicates that we quantize $z$ to a higher bitwidth. Since the step size $s_4 = 1/(2^4 - 1)$ in 4-bit quantization is a divisor of the step size $s_2$ in 2-bit quantization, the quantized values of 2-bit quantization are among the ones of 4-bit quantization. In fact, based on 2-bit quantization, the 4-bit counterpart introduces additional unshared quantized values. In particular, if $z_2$ has zero residual error, then 4-bit quantization maps $z$ to the shared quantized values (i.e., $z_2$). In contrast, if $z_2$ is with non-zero residual error, 4-bit quantization is likely to map $z$ to the unshared quantized values. In this case, 4-bit quantization can be regarded as performing quantized value re-assignment based on $z_2$. Such a re-assignment process can be formulated as follows:

$$z_4 = z_2 + \epsilon_4, \tag{9}$$

where $z_4$ is the 4-bit quantized value and $\epsilon_4$ is the re-assignment offset based on $z_2$. To ensure that the results of re-assignment fall into the unshared quantized values, the re-assignment offset $\epsilon_4$ must be an integer multiplying of the 4-bit step size $s_4$. Formally, $\epsilon_4$ can be computed by performing 4-bit quantization on the residual error of $z_2$:

$$\epsilon_4 = D(z - z_2, s_4), \quad s_4 = \frac{s_2}{2^2 + 1} = \frac{1}{2^4 - 1}. \tag{10}$$

Therefore, according to Eq. (9), a 4-bit quantized value can be decomposed into the 2-bit representation and its re-assignment offset. Similarly, an 8-bit quantized value can also be decomposed into the 4-bit representation and its corresponding re-assignment offset. In this way, we can generalize the idea of decomposition to arbitrary effective bitwidths as follows.

**Definition 1 (Quantization decomposition)** *Let $z \in [0, 1]$ be a normalized full-precision input, $\{b_1, ..., b_K\}$ be a sequence of candidate bitwidths, and $b_1 < b_2, ..., < b_{K-1} < b_K$. We use the following quantized $\widehat{z}$ to approximate $z$:*

$$\widehat{z} = z_{b_1} + \sum_{j=2}^{K} \epsilon_{b_j}, \quad \text{where } \epsilon_{b_j} = D(z - z_{b_{j-1}}, s_{b_j}), \quad s_{b_j} = \frac{s_{b_{j-1}}}{2^{b_{j-1}} + 1} = \frac{1}{2^{b_j} - 1}. \tag{11}$$

In other words, the quantized approximation $\widehat{z}$ can be decomposed into the sum of the lowest bit quantization and a series of recursive re-assignment offsets. In Definition (1), to enable quantized value re-assignment, we need to constrain that $s_{b_{j-1}}$ is divisible by $s_{b_j}$, which requires the bitwidths $b_j (j > 1)$ to satisfy the following relation:

$$b_j = 2^{j-1} \cdot b_1. \tag{12}$$

In fact, the bitwidth $b_1$ can be set to arbitrary appropriate integer values (e.g., 1, 2, 3, etc.). To get a hardware-friendly compressed network[1], we set $b_1$ to 2, which ensures that all the decomposition bitwidths are power-of-two. Moreover, since 8-bit quantization achieves lossless performance compared with the full-precision counterpart (Zhou et al., 2016), we only consider those candidate bitwidths that are not greater than 8-bit. In other words, we constrain the value of $j$ to $[1, 3]$.

**Remark 1** The proposed bit sharing decomposition has several advantages. First, the proposed method only needs to maintain a small number of trainable parameters, which greatly reduces the computational costs during search. Second, we are able to directly extract a low-precision representation from its higher precision, which allows optimizing different bitwidths jointly and ease the discontinuous optimization due to quantization.

### 3.3 LEARNING FOR COMPRESSION

Note that different layers have different levels of redundancy, which indicates that different layers may choose different subsets of the quantized values. To learn the quantized approximation for each layer, we introduce a layer-wise binary quantization gate $g_{b_j}^q \in \{0, 1\}$ on each of the re-assignment offsets in Eq. (11) to encode the choice of the quantization bitwidth, which can be formulated as

$$g_{b_j}^q = \mathbb{1}\left(||z - z_{b_{j-1}}|| - \alpha_{b_j}^q > 0\right),$$
$$\widehat{z} = z_{b_1} + g_{b_2}^q\left(\epsilon_{b_2} + \cdots + g_{b_{K-1}}^q\left(\epsilon_{b_{K-1}} + g_{b_K}^q \epsilon_{b_K}\right)\right), \tag{13}$$

---
[1]More details can be found in Appendix A.

where $\mathbb{1}(\cdot)$ is the indicator function and $\alpha_{b_j}^q$ is a layer-wise threshold that controls the choice of bitwidth. Specifically, if the quantization error $||z - z_{b_{j-1}}||$ is greater than the threshold $\alpha_{b_j}^q$, we activate the corresponding quantization gate to increase the bitwidth so that the residual error can be reduced, and vice versa.

Note that from Eq. (13), we can consider the filter pruning as 0-bit filter-wise quantization. To avoid the prohibitively large filter-wise search space, we propose to divide the filters into groups based on indexes and consider the group-wise sparsity instead. To be specific, we introduce a binary gate $g_c^p$ for each group to encode the choice of pruning, which can be formulated as follows:

$$
\begin{aligned}
g_c^p &= \mathbb{1}(||w_c|| - \alpha^p > 0), \\
\widehat{z}_c &= g_c^p \cdot \left( z_{c,b_1} + g_{b_2}^q \left( \epsilon_{c,b_2} + \cdots + g_{b_{K-1}}^q \left( \epsilon_{c,b_{K-1}} + g_{b_K}^q \epsilon_{c,b_K} \right) \right) \right),
\end{aligned}
\tag{14}
$$

where $\widehat{z}_c$ is the $c$-th group of quantized filters and $\epsilon_{c,b_j}$ is the corresponding re-assignment offset by quantizing the residual error $z_c - z_{c,b_{j-1}}$. Here, $\alpha^p$ is a layer-wise threshold for filter pruning. Following PFEC (Li et al., 2017a), we use $\ell_1$-norm to evaluate the importance of the filter. Specifically, if a group of filters is important, the corresponding pruning gate will be activated and vice versa.

Note that both quantization and pruning have their corresponding thresholds. Instead of manually setting the thresholds, we propose to learn them via gradient descent. However, the indicator function in Eq. (13) is non-differentiable. To address this, we use straight-through estimator (STE) (Bengio et al., 2013; Zhou et al., 2016) to approximate the gradient of the indicator function $\mathbb{1}(\cdot)$ using the gradient of the sigmoid function $\sigma(\cdot)$, which can be formulated as:

$$
\frac{\partial g}{\partial \alpha} = \frac{\partial \mathbb{1}\,(A - \alpha)}{\partial \alpha} \approx \frac{\partial \sigma\,(A - \alpha)}{\partial \alpha} = -\sigma\,(A - \alpha)\,(1 - \sigma(A - \alpha)),
\tag{15}
$$

where $g$ is the output of a binary gate, $\alpha \in \{\alpha^p, \alpha^q\}$ is the corresponding threshold and $A$ denotes some specific metrics (i.e., $\ell_1$-norm of the filter or the quantization error). By jointly training the binary gates and the network parameters, the pruning ratio and bitwidth of each layer can be automatically determined. However, the gradient approximation of the binary gate inevitably introduces noisy signals, which can be even more severe when we quantize both weights and activations. Thus, we propose to train the binary gates of weights and activations in an alternative manner. Specifically, when training the binary gates of weights, we fix the binary gates of activations, and vice versa.

**Search Space for Model Compression.** Given an uncompressed network with $L$ layers, we use $C_l$ to denote the number of filters at the $l$-th layer. To obtain the compressed model, we first divide the filters of each layer into groups and then search for the optimal bitwidths for the considered layer. Let $B$ be the number of filters in a group. For any layer $l$, there would be $\left\lfloor \frac{C_l}{B} \right\rfloor$ groups in total. Since we quantize both weights and activations, given $K$ candidate bitwidths, there are $K^2$ different quantization configurations for each layer. Thus, for the whole network with $L$ layers, the size of the search space $\Omega$ can be computed by

$$
|\Omega| = \prod_{l=1}^{L} \left( K^2 \times \left\lfloor \frac{C_l}{B} \right\rfloor \right).
\tag{16}
$$

Eq. (16) indicates that the search space is large enough to cover the potentially good configurations.

**Training Objective.** To design a hardware-efficient network, the objective function in Eq. (7) should reflect both the accuracy of the compressed network and its computational costs. Following (Cai et al., 2019), we train the network and architecture by minimizing following loss function:

$$
\mathcal{L}(\mathbf{W}, \alpha^p, \alpha^q) = \mathcal{L}_{ce}(\mathbf{W}, \alpha^p, \alpha^q) + \lambda \log R(\mathbf{W}, \alpha^p, \alpha^q),
\tag{17}
$$

where $\mathcal{L}_{ce}(\cdot)$ is the cross-entropy loss, $R(\cdot)$ is the computational costs of the network and $\lambda$ is a balancing hyper-parameter. Following single-path NAS (Stamoulis et al., 2019), we use a similar formulation of computational costs to preserve the differentiability of the objective function. The details of the differentiable computational loss can be found in Appendix B. Once the training is finished, we can obtain the compressed network by selecting those filters and bitwidths with activated binary gates. Then, we fine-tune the compressed network to compensate the accuracy loss.

Table 1: Comparisons of different methods on CIFAR-100. "W" and "A" represent the average quantization bitwidth of the weights and activations, respectively.

| Network | Method | BOPs (M) | BOP comp. ratio | W / A | Top-1 Acc. (%) | Top-5 Acc. (%) |
|---|---|---|---|---|---|---|
| | Full-precision | 41798.6 | 1.0 | 32.0 / 32.0 | 67.5 | 90.8 |
| | 4-bit precision | 674.6 | 62.0 | 4.0 / 4.0 | 67.8±0.3 | 90.4±0.2 |
| | DQ | 1180.0 | 35.4 | 5.3 / 6.1 | 67.7±0.6 | 90.4±0.5 |
| | HAQ | 653.4 | 64.0 | 3.7 / 4.2 | 67.7±0.1 | 90.4±0.3 |
| ResNet-20 | DNAS | 660.0 | 62.9 | 4.6 / 3.8 | 67.8±0.3 | 90.4±0.2 |
| | ABS-P (Ours) | 28586.5 | 1.5 | 32.0 / 32.0 | 67.9±0.1 | 90.7±0.2 |
| | ABS-Q (Ours) | 649.5 | 64.4 | 4.4 / 4.2 | 68.1±0.1 | 90.5±0.0 |
| | ABS (Ours) | **630.6** | **66.3** | 4.4 / 4.2 | **68.1±0.3** | **90.6±0.2** |
| | Full-precision | 128771.7 | 1.0 | 32.0 / 32.0 | 71.7 | 92.2 |
| | 4-bit precision | 2033.6 | 63.3 | 4.0 / 4.0 | 70.9±0.3 | 91.2±0.4 |
| | DQ | 2222.9 | 57.9 | 3.8 / 4.6 | 70.7±0.2 | 91.4±0.4 |
| | HAQ | 2014.9 | 63.9 | 3.3 / 4.9 | 71.2±0.1 | 91.1±0.2 |
| ResNet-56 | DNAS | 2035.7 | 65.3 | 5.3 / 3.2 | 71.2±0.2 | 91.3±0.3 |
| | ABS-P (Ours) | 87021.6 | 1.5 | 32.0 / 32.0 | 71.5±0.1 | 91.8±0.2 |
| | ABS-Q (Ours) | 1970.7 | 65.3 | 4.1 / 4.0 | 71.5±0.2 | 91.5±0.2 |
| | ABS (Ours) | **1918.8** | **67.1** | 4.2 / 4.1 | **71.6±0.1** | **91.8±0.4** |

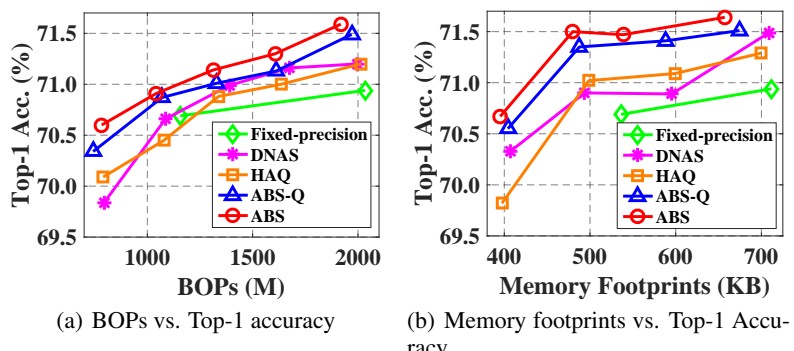

(a) BOPs vs. Top-1 accuracy

(b) Memory footprints vs. Top-1 Accuracy

Figure 2: Results of different compressed networks with different BOPs and memory footprints. We use different methods to compress ResNet-56 and report the results on CIFAR-100.

## 4 EXPERIMENTS

**Compared methods.** To investigate the effectiveness of the proposed method, we consider the following methods for comparisons: **ABS**: our proposed method with joint pruning and quantization; **ABS-Q**: ABS with quantization only; **ABS-P**: ABS with pruning only; and several state-of-the-art model compression methods including HAQ (Wang et al., 2019), DQ (Uhlich et al., 2020), DJPQ (Ying et al., 2020), Bayesian Bits (van Baalen et al., 2020) and DNAS (Wu et al., 2018). We measure the performance of different methods in terms of the Top-1 and Top-5 accuracy. Following (Guo et al., 2020; Ying et al., 2020), we measure the computational costs by the Bit-Operation (BOP) count. The BOP compression ratio is defined as the ratio between the total BOPs of the uncompressed and compressed models. We can also measure the computational costs with the total weights and activations memory footprints following DQ (Uhlich et al., 2020). Moreover, following (Stamoulis et al., 2019; Liu et al., 2019a), we use the search cost to measure the time of finding an optimal compressed model.

**Implementation details.** Following HAQ (Wang et al., 2019), we quantize all the layers, in which the first and the last layers are quantized to 8-bit. Following ThiNet (Luo et al., 2017), we only conduct filter pruning for the first layer in the residual block. For ResNet-20 and ResNet-56 on CIFAR-100 (Krizhevsky et al., 2009), we set $B$ to 4. For ResNet-18 and MobileNetV2 on ImageNet (Russakovsky et al., 2015), $B$ is set to 16 and 8, respectively. We first train the full-precision models and then use the pretrained weights to initialize the compressed models. Following (Li et al., 2020; Esser et al., 2020), we introduce weight normalization during training. We use SGD with nesterov (Nesterov, 1983) for optimization, with a momentum of 0.9. For CIFAR-100, we use the same data augmentation as in (He et al., 2016), including translation and horizontal flipping. For ImageNet, images are resized to $256 \times 256$, and then a $224 \times 224$ patch is randomly cropped from an image or its horizontal flip for training. For testing, a $224 \times 224$ center cropped is chosen. We first train the uncompressed network for 30 epochs on CIFAR-100 and 10 epochs on ImageNet.

Table 2: Comparisons on ImageNet. "*" denotes that we get the results from the figures in (van Baalen et al., 2020) and "–" denotes that the results are not reported. Moreover, "W" and "A" represent the average quantization bitwidth of the weights and activations, respectively.

| Network | Method | BOPs (G) | BOP comp. ratio | W / A | Top-1 Acc. (%) | Top-5 Acc. (%) |
|---------|--------|----------|-----------------|-------|----------------|----------------|
| ResNet-18 | Full-precision | 1857.6 | 1.0 | 32.0 / 32.0 | 70.7 | 89.8 |
| | 4-bit precision | 34.7 | 53.5 | 4.0 / 4.0 | 71.0 | 89.8 |
| | DQ | 40.7 | 40.6 | – / – | 68.5 | – |
| | DJPQ | 35.5 | 52.3 | – / – | 69.1 | – |
| | HAQ | 34.7 | 53.5 | 4.0 / 4.0 | 70.2 | 89.5 |
| | Bayesian Bits* | 35.9 | 51.7 | – / – | 69.5 | – |
| | ABS-Q (Ours) | 33.1 | 56.1 | 4.5 / 3.8 | **70.9** | **89.7** |
| | ABS (Ours) | **32.3** | **57.5** | 4.6 / 4.2 | 70.8 | 89.6 |
| MobileNetV2 | Full-precision | 308.0 | 1.0 | 32.0 / 32.0 | 71.9 | 90.3 |
| | 6-bit precision | 11.2 | 27.5 | 6.0 / 6.0 | 71.8 | 90.3 |
| | DQ | 19.6 | 1.9 | 6.8 / 8.0 | 70.4 | 89.7 |
| | HAQ | 10.8 | 28.5 | 5.61 / 6.27 | 71.2 | 90.0 |
| | Bayesian Bits* | 10.8 | 28.5 | – / – | 70.9 | – |
| | ABS-Q (Ours) | 10.9 | 28.3 | 6.8 / 6.8 | **71.8** | **90.4** |
| | ABS (Ours) | **10.8** | **28.5** | 6.1 / 7.1 | 71.7 | 90.3 |

The learning rate is set to 0.001. We then fine-tune the searched compressed network to recover the performance drop. On CIFAR-100, we train the searched network for 200 epochs with a mini-batch size of 128. The learning rate is initialized to 0.1 and is divided by 10 at 80-th and 120-th epochs. Experiments on CIFAR-100 are repeated for 5 times and we report the mean and standard deviation. For ResNet-18 on ImageNet, we finetune the searched network for 90 epochs with a mini-batch size of 256. For MobileNetV2 on ImageNet, we fine-tune for 150 epochs. For all models on ImageNet, the learning rate starts at 0.01 and decays with cosine annealing (Loshchilov & Hutter, 2017).

## 4.1 MAIN RESULTS

We apply the proposed methods to compress ResNet-20, ResNet-56 on CIFAR-100 and ResNet-18, MobileNetV2 on ImageNet. We compare the performance of different methods in Table 1 and Table 2. We also show the results of the compressed ResNet-56 with different BOPs and memory footprints in Figure 2. From the results, we can see that 4-bit quantized networks achieve lossless performance. Also, 6-bit MobileNetV2 only leads to a 0.1% performance drop on the Top-1 Accuracy. Compared with fixed-precision quantization, mixed-precision methods are able to reduce the BOPs while preserving the performance. Critically, our proposed ABS-Q outperforms the state-of-the-arts baselines with less computational costs. Specifically, ABS-Q compressed ResNet-18 outperforms the one compressed by HAQ with more BOPs reduction. More critically, our proposed ABS achieves significant improvement in terms of BOPs and memory footprints. For example, in Figure 2(a), our ABS compressed ResNet-56 model yields much fewer BOPs (395.25 vs. 536.24) but achieves comparable performance compared with the fixed-precision counterpart. Moreover, by combing pruning and quantization, ABS achieves nearly lossless performance while further reducing the computational costs of ABS-Q. For example, ABS compressed ResNet-18 reduces the BOPs by 57.5× while still outperforming the full-precision network by 0.1% in terms of the Top-1 accuracy on ImageNet.

## 4.2 FURTHER STUDIES

**Effect of the bit-sharing scheme.** To investigate the effect of the bit-sharing scheme, we apply our methods to quantize ResNet-20 and ResNet-56 with and without the bit sharing scheme on CIFAR-100. We report the testing accuracy and BOPs in Table 3. We also present the search costs and consumed GPU memory measured on a GPU device (NVIDIA TITAN Xp). It can be seen from the results that the method with the bit sharing scheme consistently outperforms the ones without the bit sharing scheme while significantly reducing the search costs and GPU memory.

**Effect of the one-stage compression.** To investigate the effect of the one-stage compression scheme (perform pruning and quantization jointly), we extend ABS to two-stage optimization, where we sequentially do filter pruning and quantization, denoted as ABS-P→ABS-Q. The results are shown in Table 4. Compared with the two-stage counterpart, ABS achieves better performance with less computational costs, which shows the superiority of the one-stage optimization. For example, ABS compressed ResNet-56 outperforms the counterpart by 0.4% on the Top-1 accuracy with less computational overhead.

Table 3: Effect of the bit-sharing scheme. We report the testing accuracy, BOPs, and search costs on CIFAR-100. The search costs are measured on a GPU device (NVIDIA TITAN Xp).

| Network | Method | Top-1 Acc. | Top-5 Acc. | BOPs (M) | Search Cost (GPU hours) | GPU Memory (GB) |
|---------|--------|-----------|-----------|----------|------------------------|-----------------|
| ResNet-20 | w/o bit sharing | 67.8±0.1 | 90.5±0.2 | 664.2 | 2.8 | 4.4 |
|  | w/ bit sharing | **68.1±0.1** | **90.5±0.0** | **649.5** | **0.8** | **1.5** |
| ResNet-56 | w/o bit sharing | 71.3±0.3 | 91.4±0.4 | 2001.1 | 8.7 | 10.9 |
|  | w/ bit sharing | **71.5±0.2** | **91.5±0.2** | **1970.7** | **1.9** | **3.0** |

Table 4: Effect of the one-stage compression. We report the results of ResNet-56 on CIFAR-100.

| Network | Method | Top-1 Acc. | Top-5 Acc. | BOPs (M) |
|---------|--------|-----------|-----------|----------|
| ResNet-56 | ABS-P → ABS-Q | 70.4±0.1 | 90.8±0.2 | 1077.7 |
|  | ABS | **70.8±0.4** | **91.2±0.1** | **1042.5** |

**Effect of the alternative training scheme.** To investigate the effect of the alternative training scheme introduced in Section 3.3, we apply our method to compress ResNet-56 using a joint training scheme and an alternative training scheme on CIFAR-100. Here, the joint training scheme denotes that we train the binary gates of weights and activations jointly. From the results of Table 5, the model trained with the alternative scheme achieves better performance than the joint one, which demonstrates the effectiveness of the alternative training scheme.

Table 5: Effect of the alternative training scheme. We report the results of ResNet-56 on CIFAR-100.

| Network | Method | Top-1 Acc. | Top-5 Acc. | BOPs (M) |
|---------|--------|-----------|-----------|----------|
| ResNet-56 | Joint | 71.3±0.2 | 91.6±0.3 | 1942.4 |
|  | Alternative | **71.6±0.1** | **91.8±0.4** | **1918.8** |

**Resource-constrained compression.** To demonstrate the effectiveness of our ABS on hardware devices, we further apply our methods to compress MobileNetV2 under the resource constraints on the BitFusion architecture (Sharma et al., 2018). Instead of using BOPs, we use the latency and energy on a simulator of the BitFusion to measure the computational costs. We report the results in Table 6. Compared with fixed-precision quantization, ABS achieves better performance with lower latency and energy. Specifically, ABS compressed MobileNetV2 with much lower latency and energy even outperforms 6-bit MobileNetV2 by 0.2% in the Top-1 accuracy.

Table 6: Resource-constrained compression on BitFusion. We evaluate the proposed ABS under the latency- and energy-constrained and report the Top-1 and Top-5 accuracy on ImageNet.

| Network | Method | Latency-constrained | | | Energy-constrained | | |
|---------|--------|------------|------------|-------------|------------|------------|-------------|
|  |  | Acc.-1 (%) | Acc.-5 (%) | Latency (ms) | Acc.-1 (%) | Acc.-5 (%) | Energy (mJ) |
| MobileNetV2 | 6-bit precision | 71.8 | 90.3 | 24.9 | 71.8 | 90.3 | 32.8 |
|  | ABS (Ours) | **72.0** | **90.4** | **17.2** | **72.0** | **90.3** | **26.3** |

## 5 CONCLUSION AND FUTURE WORK

In this paper, we have proposed a novel model compression method called Automatically Bit Sharing (ABS). Specifically, our ABS is based on the observation that quantized values of a high bitwidth share the ones of lower bitwidths under some constraints. We therefore have proposed the decomposition of quantization that encapsulates all candidate bitwidths. Starting from a low bitwidth in the search space, we sequentially increase the effective bitwidth by recursively adding re-assignment offsets. Based on this, we have further introduced learnable binary gates to encode the choice of different compression policies. By training the binary gates, the optimal compression ratio of each layer can be automatically determined. Experiments on CIFAR-100 and ImageNet have shown that our methods are able to achieve significant cost reduction while preserving the performance. In the future, we plan to work on a joint search for architecture, pruning, and quantization to find a compact model with better performance.

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

# Appendix for ABS: Automatic Bit Sharing for Model Compression

## A HARDWARE-FRIENDLY DECOMPOSITION

As mentioned in Sec. 3.2, $b_1$ can be set to arbitrary appropriate integer values (e.g., 1, 2, 3, etc.). By default, we set $b_1 = 2$ for better hardware utilization. On general purpose computing devices (e.g., CPU, GPU), $byte$ (8 bits) is the lowest data type for operations. Other data types and ALU registers are all composed with multiple bytes in width. By setting $b_1 = 2$, 2-bit/ 4-bit/ 8-bit quantization values can be packed into $byte$ (or $short, int, long$) data type without bit wasting. Otherwise, if $b_1 = 1$ or $b_1 = 3$, it is inevitable to have wasted bits when packing mixed-precision quantized tensors on general purpose devices. For example, one 32-bit $int$ data type can be used to store ten 3-bit quantized values with 2 bits wasted. One might argue that these 2 bits can be leveraged with the next group of 3-bit data, but it will result in irregular memory access patterns, which will degrade the hardware utilization more seriously. Moreover, 8-bit quantization has been demonstrated to own similar performance with the full precision counterparts for many networks. Therefore, there is no need to consider a bitwidth larger than 8.

## B FORMULATION OF DIFFERENTIABLE COMPUTATIONAL LOSS

In this section, we introduce the differentiable computational loss mentioned in Section 3.3. Unlike the cross-entropy loss in Eq. (17), the computational costs $R(\mathbf{W}, \alpha^p, \alpha^q)$ is non-differentiable. To solve this issue, we model the computational costs as a function of binary gates as:

$$R(\mathbf{W}, \alpha^p, \alpha^q) = \sum_{c=1}^{G} g_c^p \left( R_{x_{c,b_1}} + g_{b_2}^q \left( R_{x_{c,b_2}} - R_{x_{c,b_1}} + \cdots + g_{b_K}^q \left( R_{x_{c,b_K}} - R_{x_{c,b_{K-1}}} \right) \right) \right),$$

(18)

where $R_{x_{c,b_j}}$ is the computational cost for the $c$-th group of filters with $b_j$-bit quantization and $G$ is the number of groups in total.

## C QUANTIZATION CONFIGURATIONS

All the methods in Tables 1 and 2 use layer-wise and symmetric quantization schemes and the compared methods strictly follow the quantization configurations in their original papers. Specifically, for DQ (Uhlich et al., 2020), we parameterize the fixed-point quantizer using case U3 with $\boldsymbol{\theta} = [d, q_{\max}]$. We initialize the weights using a pre-trained model. The initial step size is set to $d = 2^{\lfloor \log_2(\max(|\mathbf{W}|)/(2^{b-1}-1)) \rfloor}$ for weights and $2^{-3}$ for activations. The remaining quantization parameters are set such that the initial bitwidth is 4-bit. For HAQ (Wang et al., 2019), we first truncate the weights and activations into the range of $[-v_w, v_w]$ and $[0, v_x]$, respectively. We then perform linear quantization for both weights and activations. To find more proper $v_w$ and $v_x$, we minimize the KL-divergence between the original weight distribution $\mathbf{W}$ and the quantized weight distribution $Q(\mathbf{W})$. For DNAS (Wu et al., 2018), we follow DoReFa-Net (Zhou et al., 2016) to quantize weights and follow PACT (Choi et al., 2018) to quantize activations. We initialize the learnable upper bound to 1. For DJPQ (Ying et al., 2020) and Bayesian Bits (van Baalen et al., 2020), we directly get the results from original papers. For other methods in Tables 1 and 2, we use the quantization function introduced in Section 3.1. The trainable quantization intervals $v_x$ and $v_w$ are initialized to 1.

## D SEARCH COST COMPARISONS

To evaluate the efficiency of the proposed ABS, we compare the search cost of different methods and report the results in Table 7. From the results, the search costs of the proposed ABS is much smaller than the state-of-the-art methods. Moreover, compared with ABS-Q, ABS only introduces a small amount of computational overhead, which demonstrates the efficiency of the proposed methods.

Table 7: Comparisons of the search costs on CIFAR-100. The search costs are measured on a GPU device (NVIDIA TITAN Xp).

| Network | Method | Search Cost (GPU hours) |
|---|---|---|
| ResNet-20 | HAQ | 5.8 |
| | DQ | 3.0 |
| | DNAS | 2.8 |
| | ABS-Q (Ours) | 0.8 |
| | ABS-P (Ours) | 0.2 |
| | ABS (Ours) | 1.0 |

Table 8: Comparisons of different methods w.r.t. memory footprints. We compress ResNet-56 using different methods and report the results on CIFAR-100.

| Method | Memory footprints (KB) | M.f. comp. ratio | Top-1 Acc. (%) | Top-5 Acc. (%) |
|---|---|---|---|---|
| Full-precision | 5653.4 | 1.0 | 71.7 | 92.2 |
| 4-bit precision | 711.7 | 7.9 | 70.9±0.3 | 91.2±0.4 |
| DNAS | 708.9 | 8.0 | 71.5±0.2 | 91.3±0.1 |
| HAQ | 700.0 | 8.1 | 71.3±0.1 | 91.1±0.1 |
| ABS-Q (Ours) | 674.5 | 8.4 | 71.5±0.2 | 91.6±0.2 |
| ABS (Ours) | **657.3** | **8.6** | **71.6±0.1** | **91.8±0.4** |

Table 9: Comparisons of different methods with MobileNetV3 on CIFAR-100.

| Method | BOPs (M) | BOP comp. ratio | Top-1 Acc. (%) | Top-5 Acc. (%) |
|---|---|---|---|---|
| Full-precision | 68170.1 | 1.0 | 76.1 | 93.9 |
| 6-bit precision | 2412.6 | 28.3 | 76.1±0.0 | 93.7±0.0 |
| DQ | 2136.3 | 31.9 | 75.9±0.1 | 93.7±0.1 |
| HAQ | 2191.7 | 31.1 | 76.1±0.1 | 93.5±0.0 |
| DNAS | 2051.9 | 33.2 | 76.1±0.1 | 93.7±0.1 |
| ABS-P (Ours) | 59465.8 | 1.1 | 76.0±0.0 | 93.5±0.0 |
| ABS-Q (Ours) | 2021.9 | 33.7 | 76.1±0.1 | 93.7±0.1 |
| ABS (Ours) | **2006.6** | **34.0** | **76.1±0.1** | **93.7±0.1** |

## E    MORE RESULTS ON MEMORY FOOTPRINTS

To further demonstrate the effectiveness of the proposed ABS, we replace BOPs with total weights and activations memory footprints (Uhlich et al., 2020). We apply different methods to compress ResNet-56 and report the results in Table 8. From the results, ABS compressed ResNet-56 outperforms other methods with fewer memory footprints. These results show the effectiveness of our proposed ABS in terms of memory footprints reduction.

## F    DETAILED STRUCTURE OF THE COMPRESSED NETWORK

We illustrate the pruning rate and bitwidth of each layer's weights and activations of the compressed ResNet-18 and MobileNetV2 in Figure 3 and Figure 4, respectively. From the results, we observe that our ABS assigns more bitwidths to the weights in the downsampling convolutional layer in ResNet-18 and depthwise convolutional layer in MobileNetV2. Intuitively, this is because the number of parameters of these layers is much smaller than other layers. Moreover, our ABS inclines to prune more filters in the shallower layers, which can significantly reduce the number of parameters and computational overhead. Finally, we also observe that the correlation between the bitwidth and pruning rate is as follows. If a layer is set to a high pruning rate, our ABS tends to select a higher bitwidth to compensate for the performance drop. In contrast, if a layer is with a low pruning rate, our ABS tends to select a lower bitwidth to reduce the model size and computational costs.

## G    MORE RESULTS ON MOBILENETV3

To evaluate the proposed ABS on the lightweight model, we apply our methods to MobileNetV3 on CIFAR-100. Following LSQ+ (Bhalgat et al., 2020), we introduce a learnable offset to handle the negative activations in hard-swish. We show the results in Table 9. From the results of MobileNetV3, our proposed ABS still outperforms the compared methods, which demonstrates its effectiveness.

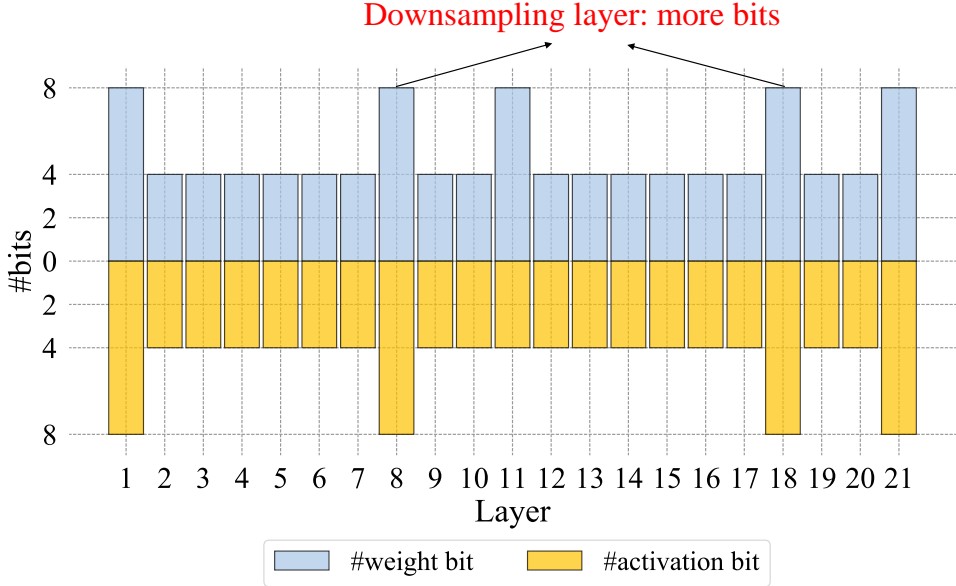

(a) Bitwidth configuration of the compressed ResNet-18.

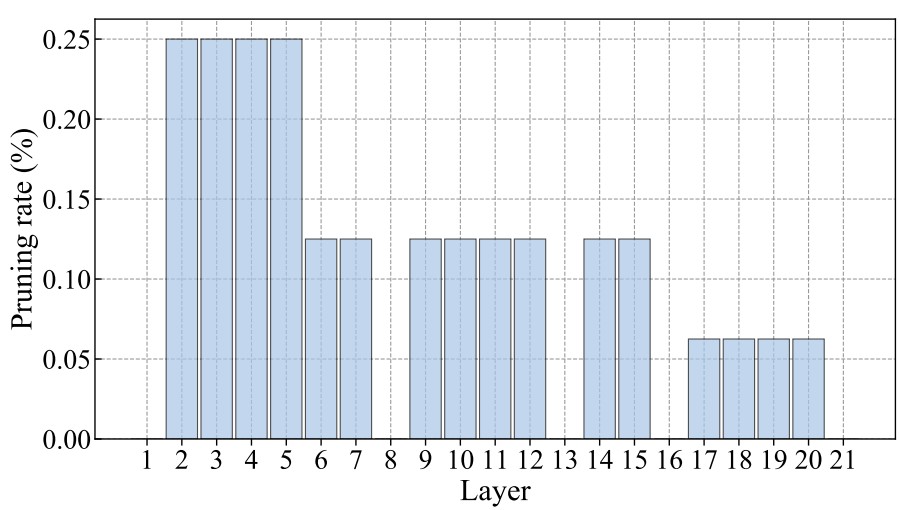

(b) Pruning rate configuration of the compressed ResNet-18. The pruning rate is defined as the ratio between #pruned weights of the compressed models and #weights of the uncompressed models.

Figure 3: Detailed configurations of the compressed ResNet-18.

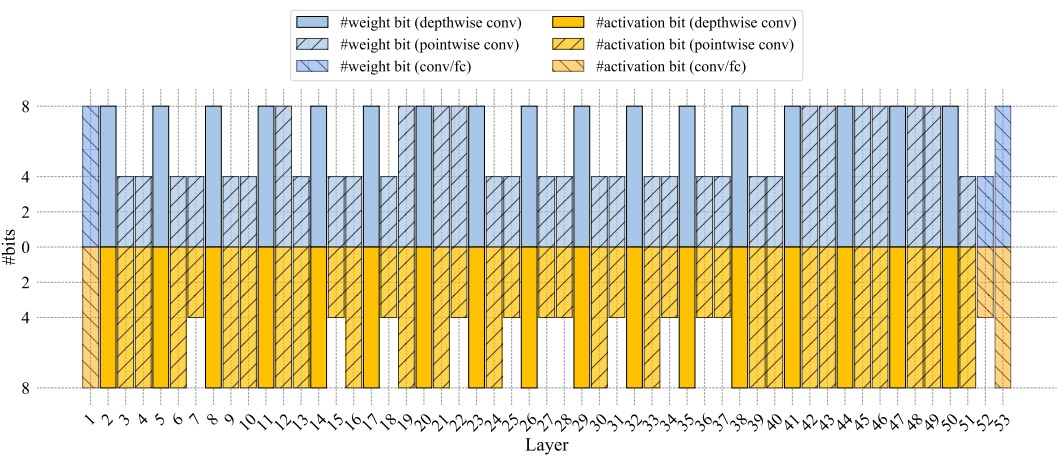

(a) Bitwidth configuration of the compressed MobileNetV2.

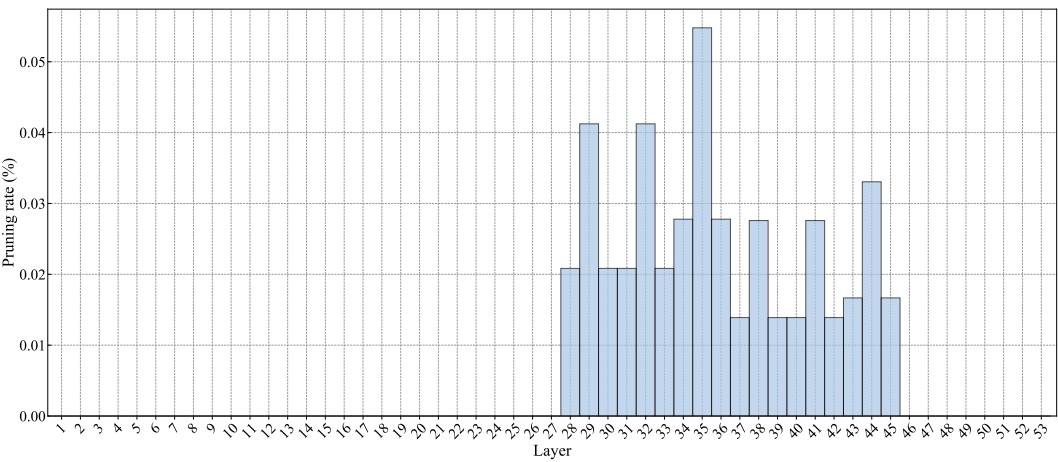

(b) Pruning rate configuration of the compressed MobileNetV2. The pruning rate is defined as the ratio between #pruned weights of the compressed models and #weights of the uncompressed models.

Figure 4: Detailed configurations of the compressed MobileNetV2.

