# OpenReview forum: "ABS: Automatic Bit Sharing for Model Compression"
_ICLR.cc/2021/Conference — Reject_

### Official Review · AnonReviewer1 · 2020-10-26
**A good paper but maybe lacks novelty**

**Rating:** 6
**Confidence:** 4

**Review:**

Summary:
	The paper presents a technique called ABS (Automatic Bit Sharing), that combines group-wise filter pruning with residual-aware quantisation in a NAS framework. The presented results on a range of image classification tasks show great performance when evaluating using an Accuracy/BOps metric. In general, I think the paper shipped interesting ideas in terms of automating residual quantisation, but it could be better if the network architecture space is jointly optimised.


Strength:
1. The paper is well-written (although it has some minor flaws), but in general easy to follow.
2. I like the formulation of using thresholds to automatically decide numerical representations, this looks neat and novel to me.
3. It is the first time for me to see residual quantisation in a NAS framework and the formulation of the indicator function looks interesting.
4. The composed search space gives a reduction in search time.

Weakness:
1. Both channel pruning and residual quantisation are existing ideas, this might reduce the novelty aspect of this paper.
2. This work focuses on searching for compression on pretrained architectures, such as ResNets and MobileNets. However, these models are not state-of-the-art on the considered datasets, and one might argue that these models are inherently more redundant than models like MobileNetV3. Also, the authors do not consider jointly optimise the network architectures, which could naturally be an extension of this work and might help to further boost the accuracy.

My suggestions & confusions:
1. I stumbled with the number representation until finishing reading section 3.1. It is probably worth mentioning that you are applying fixed-point quantisation to residual terms at the very beginning of the paper, especially in figure 1.
2. The argument that you’ve introduced 0-bit quantisation for pruning is very misleading. Apparently your pruning is channel-wise but quantisation is layer-wise, it is naturally to consider you used 0-bit quantisation on the entire layer, but this is not the case.
3. In Page 6, you mentioned you train the binary gates for weights and activations alternatively and this provides better performance. Could you quantify the performance gap? Also, if one simply looks at eq(17), this interleaved optimisation is not captured in this over-simplified optimisation target.
4. Third line of page 6, I guess you mean group-wise sparsity?
5. Page 6: Specifically, if a group of filters are ... -> if a group of filters is …?
6. You mentioned you are using group-wise sparsity, do you show the number of groups anywhere? What is your choice of B in Eq(16) for all the experiments? Or you simply mean you are doing channel-wise sparisity? I am a little confused here.

---

> ### Author Response · Authors · 2020-11-18
> **Our response to Reviewer #1**
>
> Thanks for your constructive comments and suggestions.
>
> **Q1.1.** Both channel pruning and residual quantization are existing ideas, this might reduce the novelty aspect of this paper.
>
> **A1.1.** The main novelty of this paper is to automatically search for optimal model compression configurations in a single-path framework rather than channel pruning or residual quantization. Existing compression methods (Wu et al., 2018; Dong & Yang, 2019) maintain separate paths for each configuration, which gives rise to a huge number of trainable parameters and high computational overhead. To address this, we present a super-bit that encodes all candidate configurations and introduce learnable binary gates to automatically determine the optimal configurations.
>
> We believe that our method provides a new perspective to automatic model compression. Also, the novelty of the trainable binary gates has been acknowledged by AR3 “the proposed super-bit use different gate function (forward step, backward sigmoid) and different loss function, so there is some novelty” and AR2 “the idea of presenting trainable binary quantization gate signals to automatically assign the number of quantization bits to layers is interesting”.
>
> **Q1.2.** Results on MobileNetV3.
>
> **A1.2.** Thanks for your suggestion. Due to the limited rebuttal period, results for MobileNetV3 on CIFAR-100 will be included before the end of the rebuttal. Results on ImageNet will be reported in the final version.
>
> **Q1.3.** Joint search for network architecture, pruning, and quantization.
>
> **A1.3.** Thanks for your suggestion. Our paper focuses on improving the search efficiency and the accuracy-vs-efficiency trade-off for automatic model compression rather than neural architecture design. Extension to architecture search is beyond the scope of this paper but indeed is a natural extension of our approach by directly introducing the architecture configurations (e.g., kernel size) into the search space. We have included it as a future work discussion in the conclusion.
>
> **Q1.4.** Suggestions on writing improvement.
>
> **A1.4.** Thanks for your suggestion. We have improved the mentioned Figure 1 and the corresponding descriptions in the revised paper.
>
> **Q1.5.** Confusion on 0-bit quantization.
>
> **A1.5.** Thanks for your suggestion. We use the 0-bit filter-wise quantization for pruning instead and have improved the corresponding descriptions in the revised paper.
>
> **Q1.6.** Performance improvement brought by alternative training.
>
> **A1.6.** As mentioned in Section 3.3, the gradient approximation of the binary gates inevitably introduces noisy signals, which incurs optimization difficulty during training. To solve this, we propose to train the binary gates of weights and activations in an alternative manner. From Table E, training in an alternative manner does improve performance. We have included these results in Section 4.2.
>
> Table E: Effect of the alternative training scheme. We report the results of ResNet-56 on CIFAR-100.
>
> |       Network     |    Method   | Top-1 Acc.(%) | Top-5 Acc.(%)  | BOPs (M) |
> | :------------------: | :------------: | :-----------------: | :------------------: |  :-----------: |
> |                         |      Joint      |    71.3 ± 0.2    |    91.6 ± 0.3      |   1942.4   |
> |    ResNet-56    |  Alternative |    **71.6 ± 0.1**    |   **91.8 ± 0.4**      |   **1918.8**   |
>
>
> **Q1.7.** Choice of B (the number of filters in a group).
>
> **A1.7.** As mentioned in the implementation details in Section 4, for ResNet-20 and ResNet-56 on CIFAR-100, we set B to 4. For ResNet-18 and MobileNetV2 on ImageNet, B is set to 16 and 8, respectively.
>
> **Q1.8.** Minor issues.
>
> **A1.8.** Thanks for pointing out these issues. We have revised our paper according to your suggestions.
>
> We have also summarized all of our updates in our general response. If there are any additional comments on the revised paper, please don’t hesitate to let us know.

---

> ### Author Response · Authors · 2020-11-25
> **Results on MobileNetV3**
>
> **Q1.2.** Results on MobileNetV3.
>
> **A1.2.** Thanks for your suggestion. We apply our methods to MobileNetV3 on CIFAR-100. Following LSQ+ (Bhalgat et al., 2020), we introduce a learnable offset to handle the negative activations in hard-swish. We show the results in Table D. Due to the limited rebuttal period, results on ImageNet will be reported in the final version. From the results, even on compact MobileNetV3, our proposed ABS still outperforms other methods, which demonstrates its effectiveness. We have included the results and corresponding descriptions in Section G.
>
> Table D: Comparisons of different methods with MobileNetV3 on CIFAR-100.
>
> |       Method     | BOPs (M) | BOP comp. ratio | Top-1 Acc.(%) | Top-5 Acc.(%) |
> |:-----------------:|:----------:|:---------------------:|:-----------------:|:------------------:|
> |  Full-precision |  68170.1  |            1.0            |          76.1        |         93.9         |
> | 6-bit precision |   2412.6   |           28.3           |     76.1 ± 0.0    |     93.7 ± 0.0    |
> |          DQ         |   2136.3   |           31.9           |     75.9 ± 0.1    |     93.7 ± 0.1    |
> |         HAQ        |   2191.7   |           31.1          |      76.1 ± 0.1    |     93.5 ± 0.0   |
> |        DNAS       |  2051.9    |           33.2          |     76.1 ± 0.1     |     93.7 ± 0.1    |
> |  ABS-P (Ours)  | 59465.8  |            1.1           |     76.0 ± 0.0     |     93.5 ± 0.0    |
> |  ABS-Q (Ours) |   2021.9   |          33.7           |     76.1 ± 0.1     |     93.7 ± 0.1    |
> |    ABS (Ours)   |   **2006.6**   |          **34.0**           |     **76.1 ± 0.1**     |     **93.7 ± 0.1**    |

---

### Official Review · AnonReviewer2 · 2020-10-26

**Rating:** 4
**Confidence:** 4

**Review:**

Inspired by gradient-based NAS of single-path formulation, the authors propose a super-bit model, a single-path method, to decide the optimal number of quantization bits and pruning of a group of filters. While it can be a time-consuming process to study the impact of quantization of certain filters (or layers) on model accuracy, the proposed scheme finds a particular compression configuration in a trainable manner. The experimental results show that the proposed method presents higher model accuracy or lower computational cost (measured as the bit-operation count).

While the idea of presenting trainable binary quantization gate signals to automatically assign the number of quantization bits to layers is interesting, this reviewer has the following concerns:

- For Table 1 and 2, are all methods assuming the same quantization configurations (e.g., layer-wise vs channel-wise and symmetric vs asymmatric, etc..)? Detailed quantization configuration should be included for fair comparisons.

- For Table 1 and 2, why top-1 accuracy numbers are very high for all methods? How 6-bit precision of MobileNetV2 can be 71.8% (almost no accuracy degradation)? Moreover, the accuracy of the 4-bit-quantized ResNet-18 on ImageNet is even higher than that of full-precision ResNet-18. Even so, why not decrease BOPs while lowering top-1 accuracy for 'ABS (ours)' such that the gain of ABS is maximized?

- BOPs look quite similar for many configurations (including the second row of a fixed number of quantization bits). Even though this reviewer does not believe BOPs can show performance benefits in hardware well, this reviewer cannot see distinguished advantages of ABS in Table 1 and Table 2. The gain on BOPs of ABS seems to be marginal.

- The results do not describe memory footprint savings. Can ABS provide a reduced memory footprint compared to previous ones for a target model accuracy? In Figure 2, if BOPs can be replaced with memory footprints, it would be useful to estimate the compression capability of the proposed method. In Figure 3 and 4, comparisons are missing.

- How can we optimize alpha_p and alpha_q? Is it sensitive to model accuracy? If the goal of this work is a fast exploration of compression configurations, optimizing alpha_p and alpha_q should be easy, but there is no relevant information in the manuscript (hence, Table 5 is not very reliable).

Overall, unfortunately, this reviewer cannot easily find the advantages of the proposed methods in the experiments.

---

> ### Author Response · Authors · 2020-11-18
> **Our response to Reviewer #2 (part 1)**
>
> Thanks for your constructive comments and suggestions.
>
> **Q2.1.** For Tables 1 and 2, are all methods assuming the same quantization configurations (e.g., layer-wise vs channel-wise and symmetric vs asymmetric, etc..)?
>
> **A2.1.** All the methods in Tables 1 and 2 use layer-wise and symmetric quantization schemes. Note that we obtain the results of all the compared methods using the quantization settings of their original papers. We have included the detailed quantization configurations for each method in Section C in the revised paper.
>
> **Q2.2.** For Tables 1 and 2, why top-1 accuracy numbers are very high for all methods? How can the low-precision models achieve near lossless or better performance compared with full-precision models?
>
> **A2.2.** Similar phenomenons can also be observed in the quantization literature, such as LSQ (Esser et al., 2020) and TET (Jin et al., 2019). For example, in LSQ, the 4-bit ResNet-18 even outperforms the full-precision one by 0.6% in the Top-1 accuracy. One possible reason is that model quantization acts as a regularizer to avoid the overfitting problem and improve the generalization ability of deep networks. Therefore, the low-precision models are able to achieve nearly lossless or better performance compared with the full-precision counterparts.
>
> **Q2.3.** Why not decrease BOPs while lowering top-1 accuracy for 'ABS (ours)' such that the gain of ABS is maximized?
>
> **A2.3.** Thanks for your suggestions. We have updated the results with different BOPs in Figure 2(a) in the revised paper. From the results, our ABS outperforms other methods by a large margin, especially at low BOPs settings.
>
> **Q2.4.** Marginal improvement in terms of BOPs and performance benefit on hardware.
>
> **A2.4.** Our proposed ABS achieves significant improvement in terms of BOPs, especially at low BOPs settings. For example, in Figure 2(a), our ABS compressed ResNet-56 model yields much fewer BOPs (395.25 vs. 536.24) but achieves comparable performance compared with the fixed-precision counterpart. Compared with low BOPs settings, the representational capability of the compressed models with high BOPs is higher. Hence, the improvement brought by our ABS will be smaller at high BOPs settings.
>
> To further demonstrate the effectiveness of our ABS on hardware devices, we further apply our methods to compress MobileNetV2 under the resource constraints on the BitFusion architecture (Sharma et al., 2018). The results are shown in Table B. Compared with fixed-precision quantization, ABS achieves better performance with lower latency and energy. We have included the results and corresponding descriptions in Section 4.2.
>
> Table B: Resource-constrained compression on BitFusion. We evaluate the proposed ABS under the latency- and energy-constrained and report the Top-1 and Top-5 accuracy of MobileNetV2 on ImageNet.
>
> |&nbsp;&nbsp;&nbsp;&nbsp;&nbsp;&nbsp;&nbsp;&nbsp;&nbsp;&nbsp;&nbsp; Method            |               | 6-bit precision |     ABS (Ours)     |
> |:-----------------------:|:-----------------------:|:----------------:|:-----------: |
> |                                  |  Top-1 Acc.(%)    |         71.8        |   **72.0**  |
> | Latency-constrained |  Top-5 Acc.(%)    |         90.3        |   **90.4**  |
> |                                  | Latency (ms)       |          24.9       |   **17.2**  |
> |                                  |  Top-1 Acc.(%)    |          71.8        |   **72.0**  |
> | Energy-constrained  |  Top-5 Acc.(%)    |          90.3        |   **90.3**  |
> |                                  |   Energy (mJ)      |         32.8         |   **26.3**  |

---

> ### Author Response · Authors · 2020-11-18
> **Our response to Reviewer #2 (part 2)**
>
> **Q2.5.** Results in terms of memory footprints.
>
> **A2.5.** From Table C, ABS compressed ResNet-56 outperforms other methods with fewer memory footprints. These results show the effectiveness of our proposed ABS in terms of memory footprints reduction. According to your suggestions, we add a figure to show the compression capability of our method in terms of memory footprint in Figure 2. We have included the results and the corresponding discussions in Section 4.1.
>
> Table C: Comparisons of different methods w.r.t. memory footprint. We compress ResNet-56 using different methods and report the results on CIFAR-100.
>
> |       Method      | Memory footprint (KB) | M.f. comp. ratio | Top-1 Acc.(%) |  Top-5 Acc.(%)   |
> | :------------------: |  :---------------------------: |  :----------------:  | :------------------: |  :------------------: |
> |  Full-precision  |            5653.4              |          1.0          |        71.7           |         92.2           |
> |  4-bit precision |             711.7               |          7.9          |      70.9±0.3      |        91.2±0.4     |
> |       DNAS        |             708.9               |          8.0          |       71.5±0.2     |        91.3±0.1     |
> |        HAQ         |             700.0               |          8.1          |       71.3±0.1     |        91.1±0.1     |
> | ABS-Q (Ours)  |             674.5               |          8.4          |       71.5±0.2     |        91.6±0.2     |
> |   ABS (Ours)    |             **657.3**               |         **8.6**           |       **71.6±0.1**     |        **91.8±0.4**     |
>
> **Q2.6.** How can we optimize alpha^p and alpha^q? Is it sensitive to model accuracy? If the goal of this work is a fast exploration of compression configurations, optimizing alpha_p and alpha_q should be easy, but there is no relevant information in the manuscript (hence, Table 7 is not very reliable).
>
> **A2.6.** We need to clarify that $\alpha^p$ and $\alpha^q$ are two learnable parameters rather than the predefined hyperparameters. Thus, we can not directly investigate the sensitivity of these parameters. To better understand Eq. (13) and Eq. (14), we equivalently change the formulation by moving $\alpha^q$ and $\alpha^p$ from the right to the left of the inequality operator. Note that the indicator function is non-differentiable. To address this, we use the straight-through estimator (STE) (Bengio et al., 2013; Zhou et al., 2016) to estimate the gradients of the indicator function. Therefore, we are able to learn $\alpha^p $ and $\alpha^q$ in conjunction with other network parameters via stochastic gradient descent. Combined with the single-path scheme, our ABS shows promising search efficiency (See Table 7 in the revised paper). We have made it clearer in the revised paper.
>
> We have also summarized all of our updates in our general response. If there are any additional comments on the revised paper, please don’t hesitate to let us know.

---

### Official Review · AnonReviewer3 · 2020-10-26
**Good work, but the novelty might be questionable**

**Rating:** 6
**Confidence:** 3

**Review:**

The paper describes the method to determine optimal quantization bit-width and pruning configuration for the neural network compression. Different from other approaches, the proposed method integrates multiple bit configurations (including pruning) into a single architecture, which is named “Super-bit”. The architecture uses binary gates to automatically select bit resolution. In addition, the super-bit model is differentiable and jointly trainable with parameters.

The idea of decomposing bit configuration into 2, 4, 8 bits, and using binary gates does not seems to be novel [van Baalen, 2020]. This paper also adopts the trainable gate parameter and unifies the pruning scheme.

Compared to [van Baalen, 2020], the proposed super-bit use different gate function (forward step, backward sigmoid) and different loss function, so there is some novelty. However, there is no direct comparison of the algorithm. To be fair, [van Baalen, 2020] is quite recent work and both ideas may be developed concurrently. So, I think inserting a single paragraph of comparison would be enough.

Using cost function R is a good idea to directly reduce the computation cost. Experiments, especially ABS-P and ABS-Q are well designed and suitable.

Some minor questions/suggestions:
1)	Does the cost R consider both weight and activation bits? If yes, the input resolution change should affect BOPs and will change the optimal bit-width. It would be meaningful to show the correlation between optimal bit-width and the number of filters (but not necessary).
2)	How is the ‘search cost’ calculated? As mentioned, the key point of the algorithm is to simultaneously train weights and configurations. Is ‘search time’ means the fine-tuning process after training the uncompressed network? Is it equal to other works? (Table 5)
3)	In Figure 3 & 4, there is no layer assigned to 2-bit. Can you provide average bit-width for ABS networks?

---

> ### Author Response · Authors · 2020-11-18
> **Our response to Reviewer #3**
>
> Thanks for your constructive comments and suggestions.
>
> **Q3.1.** Discussions with concurrent work (van Baalen, 2020).
>
> **A3.1.** Our ABS and Bayesian Bits (van Baalen, 2020) are developed concurrently that share a similar idea of quantization decomposition. Critically, our ABS differs from Bayesian Bits in several aspects:
>
> 1) The quantization decomposition in our methods can be extended to non-power-of-two bit widths (i.e., $b_1$ can be set to arbitrary appropriate integer values), which is a general case of the one in Bayesian Bits.
> 2) The optimization problems are different. Specifically, we formulate model compression as a single-path subset selection problem while Bayesian Bits casts the optimization of the binary gates to a variational inference problem that requires more relaxations and hyperparameters.
> 3) Our compressed models with less or comparable BOPs outperform those of Bayesian Bits by a large margin on ImageNet (See Table A).
>
> We have included the discussions in Section 2 and the corresponding results in Table 2 of the revised paper.
>
> Table A: Comparisons of Bayesian Bits and ABS on ImageNet. “*” denotes that we get the results from the figures in (van Baalen, 2020).
>
> |      Network     |        Method      | Top-1 Acc.(%)  | BOPs(G) |
> |  :--------------:   | :-------------------: | :-----------------:  | :----------: |
> |                        |   Full-precision  |        70.7          |   1857.6   |
> |   ResNet-18    | Bayesian Bits*  |        69.5          |    35.9      |
> |                        |    ABS (Ours)    |        **70.8**          |     **32.3**     |
> |                        |   Full-precision  |        71.9           |    308.0   |
> |  MobileNetV2  | Bayesian Bits*  |        70.9           |     10.8    |
> |                        |    ABS (Ours)    |        **71.7**           |     **10.8**    |
>
> **Q3.2.** Does the cost $R$ consider both weight and activation bits? If yes, the input resolution change should affect BOPs and will change the optimal bit-width.
>
> **A3.2.** Yes, the computational costs $R(\cdot)$ consider the bitwidths of both weights and activations. Specifically, we use Bit-Operation (BOP) count to measure the computational costs following (Guo et al., 2020; Ying et al., 2020). We agree that the change of input resolution will affect BOPs and change the optimal bitwidth. However, in our paper, we mainly focus on searching for optimal model compression configurations and only consider a fixed input resolution to exclude the effect of input resolution. We will incorporate the input resolution into the search space to investigate its effect in future work.
>
> **Q3.3.** The correlation between the bitwidth and pruning rate.
>
> **A3.3.** We have shown the correlation between the bitwidth and pruning rate in Figures 3 and 4. If a layer is set to a high pruning rate, our ABS tends to select a higher bitwidth to compensate for the performance drop. In contrast, if a layer is with a low pruning rate, our ABS tends to select a lower bitwidth to reduce the model size and computational costs. We have included the discussions in Section F.
>
> **Q3.4.** How is the ‘search cost’ calculated? Does 'search time’ mean the fine-tuning process after training the uncompressed network? Is it equal to other works? (Table 7)
>
> **A3.4.** According to the definitions in (Stamoulis et al., 2019; Liu et al., 2019a), the standard search cost refers to the time of finding an optimal architecture or compressed model. Following these papers, we compute the search cost to represent the time of training both the model weights and compression configurations. Thus, the cost of the fine-tuning does not belong to the search cost. It is worth noting that we compute the search cost of all the methods in Table 7 in the same way.
>
> **Q3.5.** Results in terms of average bitwidth.
>
> **A3.5.** Thanks for your suggestion. We have provided the results of the average bitwidth in Tables 1 and 2 in the revised paper.
>
> We have also summarized all of our updates in our general response. If there are any additional comments on the revised paper, please don’t hesitate to let us know.

---

### Author Response · Authors · 2020-11-18
**General Response**

We sincerely thank all reviewers for their valuable comments. We have revised our submission and summarized our updates as follows:

1. We have improved the writing and updated Figure 1 according to the suggestions.
2. We have added discussions with the concurrent work (van Baalen, 2020) in Section 2 and included comparison results in Table 2.
3. We have provided the results in terms of average bitwidth in Tables 1 and 2.
4. We have included more results with lower BOPs in Figure 2(a).
5. We have included the results in terms of memory footprint in Figure 2(b).
6. We have provided more results in terms of resource-constrained compression on the BitFusion platform in Section 4.2.
7. We have included the results in terms of performance improvement brought by alternative training in Section 4.2.
8. We have provided detailed quantization configurations of different methods in Section C.
9. We have provided more analyses in terms of the relationship between bitwidth and pruning rate in Section F.
10. We have included the results in terms of MobileNetV3 in Section G.

If there are any additional comments on the paper, please don’t hesitate to let us know.

---

### Decision · Program_Chairs · 2021-01-07
**Final Decision**

**Decision:**

Reject

**Comment:**

The paper proposes to integrate multiple bit configurations (including pruning) into a single architecture, and then automatically select bit resolution through binary gates. The overall approach can be differentiable and optimized with parameters. However, as pointed out by the reviewers, the novelty of this paper can be the big question. Also, it seems to be hard or unpractical to employ different number of bits for layers, given the standard GPU and CPU hardware.